# Cohesin SMC1β promotes closed chromatin and controls TERRA expression at spermatocyte telomeres

Uddipta Biswas[1],*, Tanaya Deb Mallik[1],*, Johannes Pschirer[1], Matthias Lesche[2], Katrin Sameith[2], Rolf Jessberger[1]

**Previous data showed that meiotic cohesin SMC1β protects spermatocyte telomeres from damage. The underlying reason, however, remained unknown as the expressions of telomerase and shelterin components were normal in Smc1β$^{-/-}$ spermatocytes. Here. we report that SMC1β restricts expression of the long noncoding RNA TERRA (telomeric repeat containing RNA) in spermatocytes. In somatic cell lines increased TERRA was reported to cause telomere damage through altering telomere chromatin structure. In Smc1β$^{-/-}$ spermatocytes, we observed strongly increased levels of TERRA which accumulate on damaged chromosomal ends, where enhanced R-loop formation was found. This suggested a more open chromatin configuration near telomeres in Smc1β$^{-/-}$ spermatocytes, which was confirmed by ATAC-seq. Telomere-distal regions were not affected by the absence of SMC1β but RNA-seq revealed increased transcriptional activity in telomere-proximal regions. Thus, SMC1β promotes closed chromatin specifically near telomeres and limits TERRA expression in spermatocytes.**

## Introduction

Cohesin is essential for sister chromatid cohesion and critically involved in DNA recombination and repair, regulation of gene expression, telomere protection, and possibly other processes (for reviews see [Nasmyth, 2011; Haering & Jessberger, 2012; Peters & Nishiyama, 2012; Seitan & Merkenschlager, 2012; McNicoll et al, 2013; Remeseiro & Losada, 2013; Rankin, 2015; Ishiguro, 2019; Higashi & Uhlmann, 2022; Horsfield, 2022]). In mammalian somatic cells, the cohesin complex consists of two SMC proteins, SMC1α and SMC3, which heterodimerize, and the kleisin RAD21/SCC1, which closes the cohesin ring-like structure. A fourth subunit, an SA protein, associates with the tripartite cohesin core complex. Cohesin proteins play central roles in meiosis: no gametes develop without functional cohesin. When cells enter meiosis, meiosis-specific cohesin subunits become expressed and incorporated into cohesin complexes. Although there are only two types of mitotic cohesin complexes,

distinguished by their association with either SA1 or SA2, meiocytes produce a larger number of different complexes. All of these complexes contain SMC3, but there are variants for each of the three other subunits. Three distinct kleisins, RAD21, REC8, and RAD21L, and three SA-type proteins (SA1, SA2, and SA3, also called STAG1, STAG2, and STAG3) exist in meiocytes, and there is a variant of SMC1 called SMC1β. SMC1β instead of SMC1α forms meiotic cohesin complexes throughout meioses I and II (Revenkova et al, 2001, 2004), but in prophase I, SMC1α cohesin complexes also exist in meiocytes. SMC1β is expressed only after the cells have entered meiosis (Eijpe et al, 2003) and is required for proper length and loop-axis structure of axial elements (AEs) and synaptonemal complexes (SCs), the condensed hallmark chromosome structures present in leptotene/zygotene and pachytene meiocytes, respectively.

Earlier, we reported telomere damage in SMC1β deficient spermatocytes and oocytes (Adelfalk et al, 2009). Zygotene and pachytene meiocytes showed AEs and SCs that lack telomeres on at least one end of many chromosomes, featured short separated fragments of chromosome axes carrying telomere sequences, showed much extended telomere sequences at the ends of some chromosomes, showed apparent telomere fusions and even ring-like chromosomes with an internal telomere signal. Overall, the length of intact telomeres was lower in these cohesin-deficient germ cells. In a follow-up study, it became clear that only SMC1β but not SMC1α cohesin complexes protect telomeres from such damage (Biswas et al, 2018). Although SMC1α expression driven by the SMC1β promoter in SMC1β-deficient mice (the Smc1β$^{-/-1α}$ strain) rescues most of the phenotypes associated with SMC1β deficiency such as reduced AE/SC length, asynapsis, and delayed DSB repair, it cannot rescue telomere damage. Thus, a specific function for SMC1β in mammalian meiocytes is to maintain telomere integrity.

Which activity of SMC1β may be responsible for telomere protection? In Smc1β$^{-/-}$ spermatocytes, telomerase activity appeared normal and all shelterin proteins that were assessed, that is, RAP1, TRF1, TRF2, and POT1, were present and localized properly to telomeric sequences, including to aberrant telomeres (Adelfalk et al, 2009). Therefore, we investigated another key component of telomere biology, the long noncoding RNA TERRA.

[1]Institute of Physiological Chemistry, Faculty of Medicine Carl Gustav Carus, Technische Universität Dresden, Dresden, Germany   [2]Center for Molecular and Cellular Bioengineering, Genome Center Technology Platform, Dresden, Germany

Correspondence: rolf.jessberger@tu-dresden.de
*Uddipta Biswas and Tanaya Deb Mallik contributed equally to this work

TERRA, which was identified in 2007 (Azzalin et al, 2007), stands for "telomeric repeat containing RNA," and is expressed from promoters in the subtelomeric region, running hundreds of nucleotides into the telomere repeats (for reviews see [Oliva-Rico & Herrera, 2017; Diman & Decottignies, 2018; Toubiana & Selig, 2018]). TERRA is expressed in many species ranging from yeast to man and in most cases is a product of RNA polymerase II transcription with a fraction of TERRA becoming poly-adenylated (Schoeftner & Blasco, 2008). In mammals, TERRA is best described in human cells such as established cell lines. Human TERRA promoters feature CpG islands, and the chromatin organizing factor CTCF, which often colocalizes with cohesin (Parelho et al, 2008; Rubio et al, 2008; Stedman et al, 2008; Wendt et al, 2008), was shown by ChIP-Seq to localize to most TERRA promoter regions in human cells (Nergadze et al, 2009; Deng et al, 2010, 2012a). Much less was reported on the respective mouse promoters and on features of murine TERRA. Generally, properly expressed TERRA has a telomere protective role, but if there is too little or too much of TERRA, telomere damage results (Azzalin et al, 2007; Deng et al, 2012b; Pfeiffer & Lingner, 2012). TERRA has been scarcely described for mammalian germ cells, but is present throughout prophase I in male and female meiocytes (reviewed in Reig-Viader et al [2016]). The levels of TERRA expression during spermatogenesis increase starting at the entry into meiosis, peak in meiosis II, and decrease at the onset of spermiogenesis (Reig-Viader et al, 2014).

Here, we report that expression of TERRA is negatively controlled by SMC1β, which supports formation of compacted subtelomeric chromatin, and we propose this to be important for maintaining spermatocyte telomere integrity.

## Results

### TERRA expression is strongly increased in SMC1β-deficient spermatocytes

Which telomere-relevant process is controlled by SMC1β? Because earlier, we had ruled out a role of SMC1β in expression and localization of shelterin components, telomerase expression, and telomerase activity (Adelfalk et al, 2009; Biswas et al, 2018), we analyzed expression of the long noncoding RNA TERRA, for irregular TERRA expression has been associated with chromosome rearrangements and damage (Azzalin et al, 2007; Deng et al, 2012b; Pfeiffer & Lingner, 2012).

TERRA has been described for human fetal oocytes (Reig-Viader et al, 2013) and for mouse spermatocytes (Reig-Viader et al, 2014). To assess its relative levels of expression compared with somatic tissue, we assayed TERRA expression in wild-type (wt) mouse testis compared with the spleen (Fig S1A). Quantitative RT–PCR was performed on oligo(dT)-selected RNA using chromosome-specific primers diagnostic for the subtelomeric to telomeric regions. For all four selected chromosomes representing long, medium length, and short mouse chromosomes, up to 10fold higher levels of TERRA were observed in testis than in the spleen. Control transcripts, the meiosis-specific *Spo11*, and somatic *Swap70* and *Def6*; the latter two were strongly expressed in B lymphocytes, showing the expected

patterns. To determine whether TERRA expression is altered in SMC1β-deficient mice, we investigated both, the presence of total TERRA transcripts and that of poly(A)-TERRA transcripts in mouse testis (Fig 1A). We included testis RNA from the $Smc1β^{-/-}$ and the $Smc1β^{-/-1a}$ mouse strains next to controls. With respect to telomere damage, there is no difference between $Smc1β^{-/-1a}$ and $Smc1β^{-/-}$ spermatocytes (Biswas et al, 2018). In both mutant strains, TERRA expression was much increased for both types of RNA preparations–up to 35fold–on all four chromosomes tested. The levels of increase varied between chromosomes and was generally a bit higher in the $Smc1β^{-/-1a}$ mice. This may be because of spermatogenesis proceeding slightly further up to early diplotene in the $Smc1β^{-/-1a}$ strain, which may allow accumulation of more TERRA transcripts.

Next, we FACS-sorted 4N spermatocytes (FigS1B) from control and SMC1β-deficient testes to specifically analyze TERRA expression in prophase I spermatocytes (Fig 1B). In addition to the four representative chromosomes used before, we included primers for chromosome 3 and for the pseudoautosomal region (PAR) of the X/Y chromosomes. The PAR (Solari, 1970) is ~700 MB long, synapses later than the autosomes, repairs the programmed DSBs later, and like the entire X/Y chromosomes is embedded in the sex body chromatin, which is transcriptionally repressed and features respective marks such as SUMO-1 and γH2AX. Strongly increased levels of poly(A)-TERRA in SMC1β deficient 4N spermatocytes were found on all five chromosomes. The increase ranges from 2.5 to 60fold on the autosomes and is 25fold on the PAR (Fig 1B).

We also analyzed pachytene oocytes from wt and $Smc1β^{-/-}$ mice isolated at embryonic days 17.5–18.5, that is, in their pachytene stage. Although there seemed to be a trend towards higher TERRA expression in all cases, only one of the five chromosomes analyzed showed a statistically significant increase of about twofold in TERRA expression (Fig S2A). Investigation of oocytes from young adult mice, that is, oocytes in dictyate arrest, again revealed increased TERRA expression for only one of five chromosomes analyzed (Fig S2B). Thus, there is no or only a small difference between wt and $Smc1β^{-/-}$ TERRA expression in oocytes at these two stages of oogenesis.

Using an RNA-FISH protocol, we assessed the presence of TERRA at the ends of chromosomes in chromosome spreads of wt, $Smc1β^{-/-1a}$, and $Smc1β^{-/-}$ early pachytene spermatocytes (Fig 2A). Quantification of the number of TERRA signals showed an increase in the mutant spermatocytes (Fig 2B). As expected, $Smc1β^{-/-1a}$ and $Smc1β^{-/-}$ spermatocytes were not different. Likewise, the total TERRA signal intensity per cell was increased in the two mutant strains. These data reflect the increase in TERRA expression and TERRA localizing to chromosome ends. In the mutants, TERRA also associated with their extended telomere signals as these cells show a few stretches of telomeres and additional chromosome fragments carrying telomeres (Adelfalk et al, 2009; Biswas et al, 2018).

Next, we compared within the $Smc1β^{-/-1a}$ mutant the undamaged with the damaged ends of individual chromosomes for the presence of TERRA, assuming that the damaged ends may show increased presence of TERRA (Fig 3). The relative signal intensity of RNA FISH (TERRA) and of DNA FISH (all telomere sequences) within $Smc1β^{-/-1a}$ spermatocytes was compared. The same type of staining for the wt, which does not feature damaged chromosome ends, is

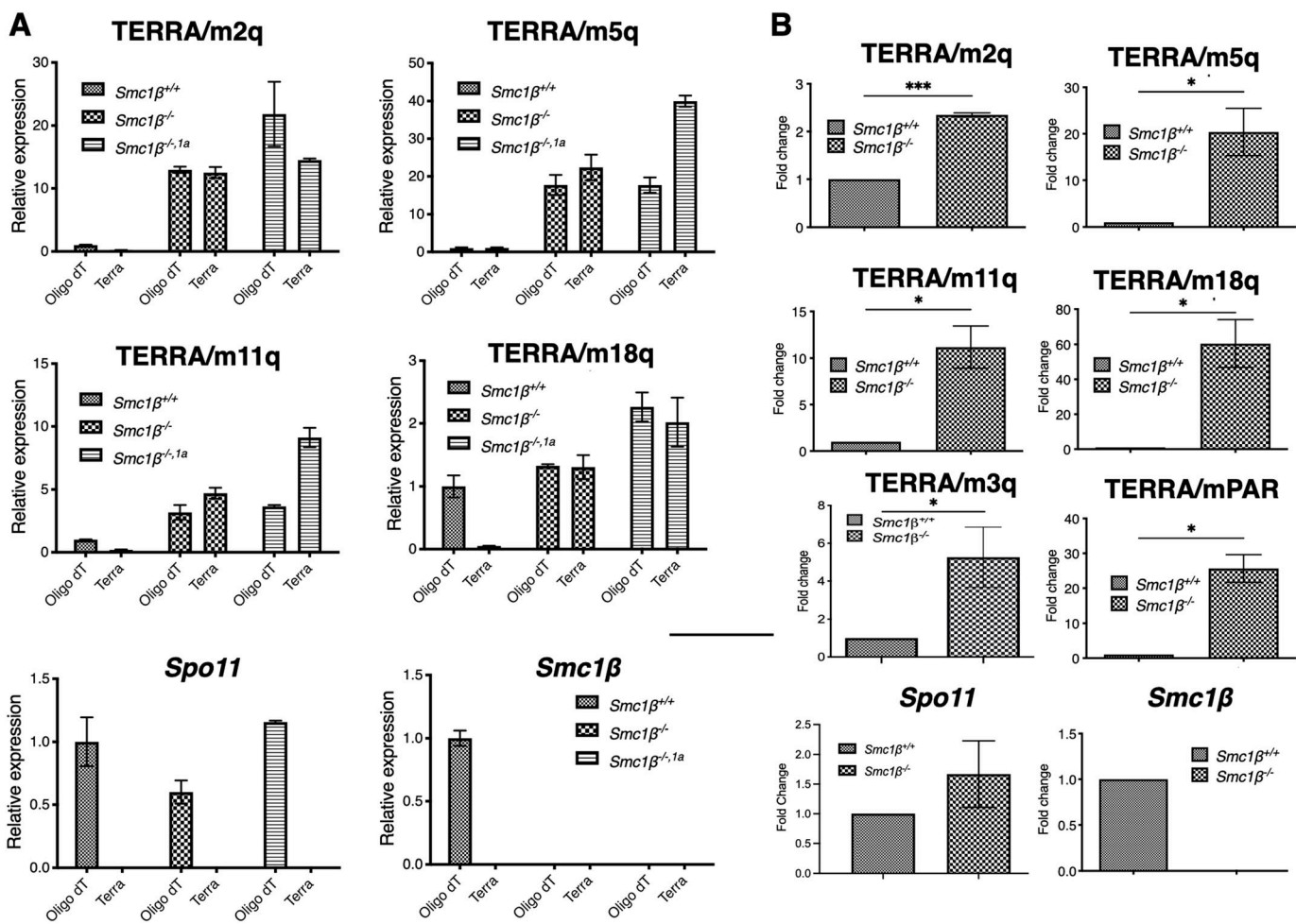

**Figure 1. Expression levels of TERRA in control and SMC1β-deficient spermatocytes.**
**(A)** Graphs showing the relative expression of TERRA in *Smc1β*[+/+], *Smc1β*[−/−], and *Smc1β*[−/−,1a] spermatocytes in whole testis (n = 6 independent experiments from three different mice each genotype). cDNA was prepared either using oligo dT or TERRA-specific oligonucleotides. **(B)** Graphs showing the relative expression of TERRA in *Smc1β*[+/+] and *Smc1β*[−/−]-sorted spermatocytes (n = 3 independent experiments, three mice each genotype, each from litter mates). cDNA was prepared by using oligo dT primers. The β-actin expression was used to normalize expression. Unpaired *t* test was performed for statistical significance.
Source data are available for this figure.

illustrated in Fig S3. There was a 3.5fold higher total TERRA RNA signal intensity on the damaged ends of chromosomes compared with the undamaged ends. The increase in telomeric DNA signal on damaged versus non-damaged chromosome ends was 2.7fold (Fig 3C). Both values are statistically significant ($P$ = 0.0037 and 0.036, resp.), with a 10fold lower $P$-value for the increase in TERRA. More telomeric DNA indicates aberrant, damaged telomers including stretches, bridges, separated telomeric fragments, etc. With extended, damaged chromosome ends, more TERRA can associate. The increase in TERRA is even slightly higher than that of telomeric DNA. This ratio of 1.3fold more TERRA (RNA) signal than telomer DNA signal (3.476:2.740) was similarly reflected in measurements of the signal area (1.2fold), and the mean signal intensity per chromosome end was also slightly increased (1.1fold). These differences, because of the large variation in the type and extent of telomere damage, were not statistically significant; there was a trend towards a proportionally higher increase of TERRA than of telomere DNA at sites of damaged ends of chromosomes in the *Smc1β*[−/−1a]

spermatocytes, consistent with the accumulation of TERRA at such damaged ends.

### Increased RNA–DNA hybrids at chromosome ends in SMC1β-deficient spermatocytes

The G-rich TERRA sequences are well suited to form DNA–RNA hybrid structures, R-loops, which have been described in several organisms including yeast and mammals. Often, such R-loops were identified in cancer cells that undergo alternative lengthening of telomeres (ALT), which uses recombination pathways to maintain telomere length. Thus, a link of R-loops to genome instability at chromosome ends was described (reviewed in Aguilera and Garcia-Muse [2012]; Skourti-Stathaki and Proudfoot [2014]; Costantino and Koshland [2015]; Richard and Manley [2017]; Toubiana and Selig [2018]). Generally, DNA–RNA hybrid structures contribute to open, more accessible chromatin and are more prone to DNA damage.

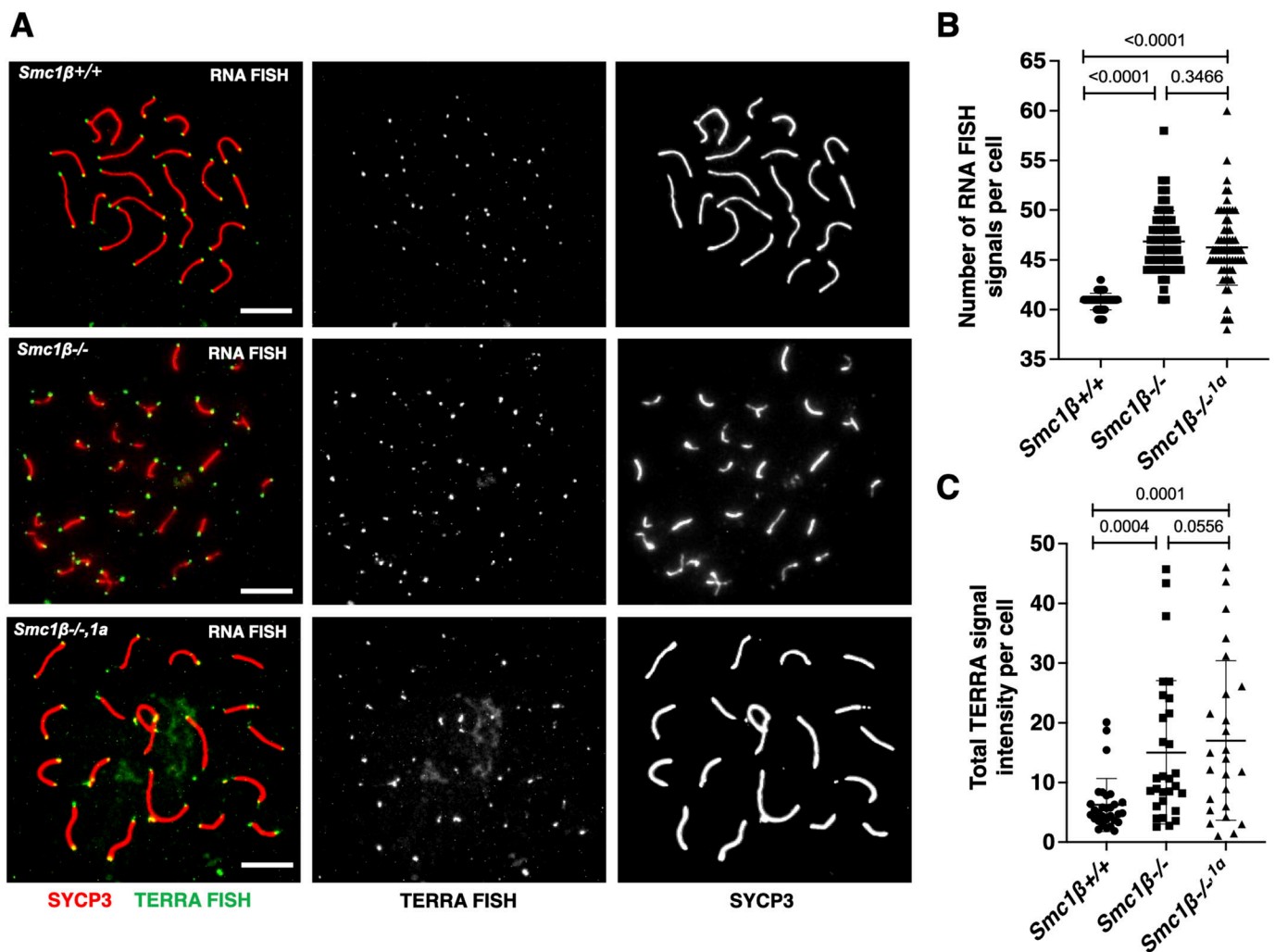

**Figure 2. Increased localization of TERRA RNA to telomeres in mutant spermatocytes.**
**(A)** RNA FISH was performed in $Smc1\beta^{+/+}$ (n = 30 cells), $Smc1\beta^{-/-}$ (n = 25 cells) and $Smc1\beta^{-/-,1a}$ (n = 40 cells) spermatocyte spreads. TERRA RNA FISH and AEs (SYCP3) are shown in green and red, respectively. Scale bar = 5 $\mu$m. **(B)** Graph showing the number of TERRA RNA FISH signals with SD in spermatocytes from $Smc1\beta^{+/+}$ (mean: 40.81; n = 64 cells from two mice), $Smc1\beta^{-/-}$ (mean: 46.83; n = 63 cells from two mice) and $Smc1\beta^{-/-,1a}$ (mean: 46.25; n = 65 cells from two mice) mice. Unpaired $t$ test was performed. **(C)** Total TERRA signal intensity for each spermatocyte was calculated per cell area. The graph shows the mean intensity with SD for $Smc1\beta^{+/+}$ (n = 31 cells), $Smc1\beta^{-/-}$ (n = 30 cells), and $Smc1\beta^{-/-,1a}$ (n = 29 cells) spermatocytes from two mice each. Unpaired $t$ test was performed.
Source data are available for this figure.

Because TERRA transcripts are known in cell lines to form RNA–DNA hybrids, which render the respective chromosomal regions non-stable (Aguilera & Garcia-Muse, 2012; Skourti-Stathaki & Proudfoot, 2014; Costantino & Koshland, 2015; Richard & Manley, 2017), we asked whether the increase in TERRA causes an increase in RNA–DNA hybrids, particularly at the spermatocytes' chromosome ends. The S9.6 antibody recognizes RNA–DNA hybrid structures and has been widely used to identify and characterize them (Boguslawski et al, 1986; Phillips et al, 2013). However, as recently shown, this antibody also recognizes rRNA (Smolka et al, 2021). RNA–DNA hybrids and rRNA can be distinguished by use of distinct RNases, RNaseH, and RNaseT1, respectively (Smolka et al, 2021). Probing wt pachytene spermatocyte chromosome spreads with S9.6 showed very little staining along chromosome axes and at the ends of chromosomes but strong staining of the sex body chromatin (Fig

4A). This typical chromatin cloud surrounding the sex chromosomes was observed because S9.6 recognized asynapsed regions of chromosomes supposed to harbour RNA–DNA hybrids. This cloud is not present in $Smc1\beta^{-/-}$ spermatocytes, but quite extensive asynapsis of autosomes is observed as reported before (Revenkova et al, 2004; Biswas et al, 2016). In addition, there was about a 10fold increase in the number of S9.6 signals at the ends of $Smc1\beta^{-/-1a}$ chromosomes indicating strongly enhanced presence of RNA–DNA hybrids at telomeres/subtelomeres in the absence of SMC1$\beta$ (Fig 4B). Asynapsis is not seen in $Smc1\beta^{-/-1a}$ spermatocytes and the S9.6 staining is restricted to the sex body and these many chromosome ends. Thus, $Smc1\beta^{-/-1a}$ spermatocytes allow analysis unperturbed by autosomal asynapsis. RNaseH treatment of such chromosome spreads was performed to confirm that the signals are indeed RNA–DNA hybrids. This treatment on fixed spreads is difficult and

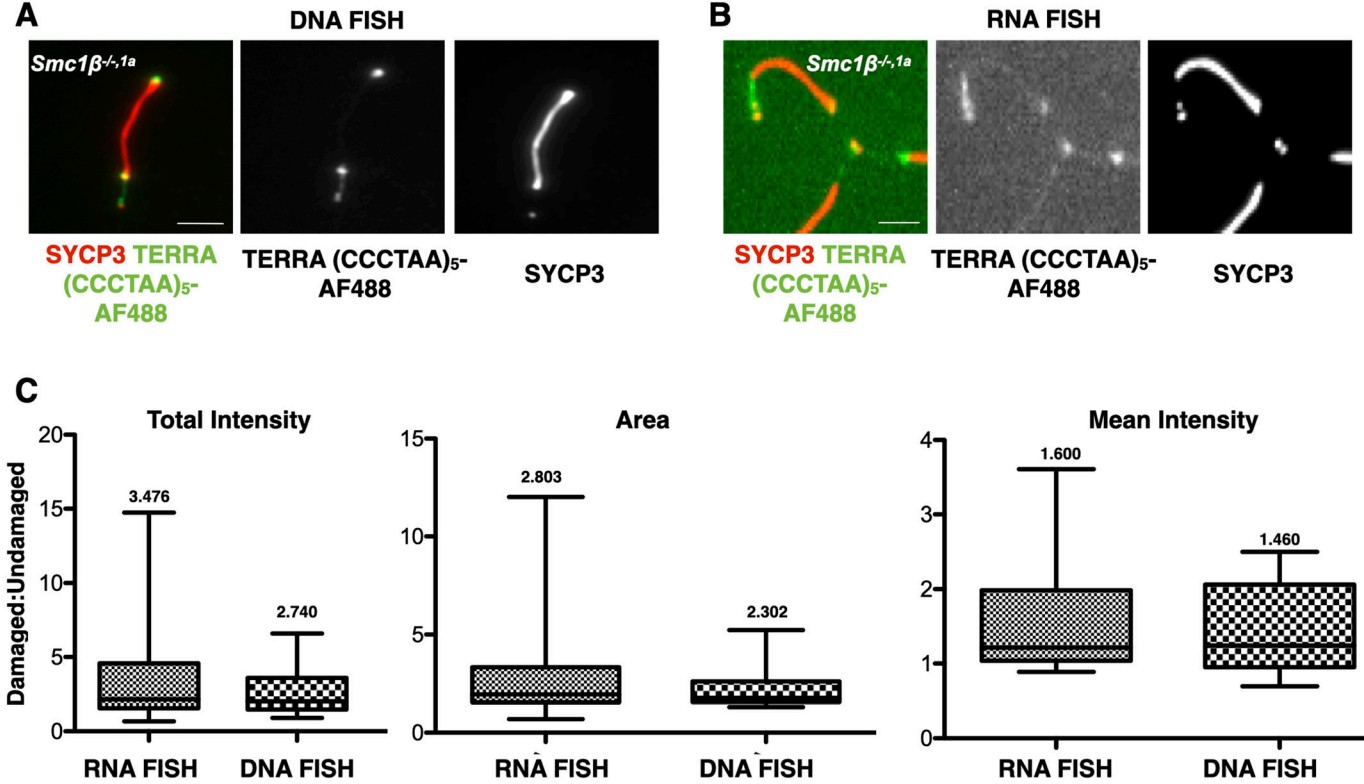

**Figure 3. TERRA signals at damaged versus non-damaged ends of chromosomes.**
**(A, B)** $Smc1\beta^{-/-,1a}$ spermatocyte spreads showing telomeric abnormalities were probed either by DNA ((A), 45 cells) or RNA FISH ((B), 56 cells) using TERRA-specific probes. The AEs were stained by anti-SYCP3. Non-axial or broken chromatid signals were assigned as damaged chromatids, which show signals for telomere and SYCP3, but are solitary, not part of the main AE. The other ends of the same chromosomes which do not show such features are assigned as non-damaged. Undamaged and damaged telomers within the $Smc1\beta^{-/-,1a}$ spermatocytes were compared. Scale bar = 5 $\mu$m. **(C)** Graphs showing the total signal intensity, area, and mean signal intensity of the ratio between damaged and non-damaged chromosome ends of $Smc1\beta^{-/-,1a}$ spermatocytes for DNA and DNA FISH signals. Statistical evaluation was performed using Dunn's multiple comparison test and the Mann–Whitney test.
Source data are available for this figure.

only a fraction of the R-loops appeared to be accessible to the enzyme. Still, the S9.6 signals were significantly reduced by ~threefold (Fig 4B). To exclude the S9.6 signals originated from rRNA, we treated the spreads simultaneously or separately with RNaseT1. RNaseT1 alone slightly reduced the S9.6 signals, but single RNaseH or combined RNaseH/T1 reduced the signals much more. This shows that a large portion of the S9.6 signals are indeed RNA–DNA hybrids. Thus, most of the S9.6 signals on the spreads represented R-loops. Autosomes and sex chromosomes showed similarly reduced S9.6 signals upon RNaseH treatment, but little reduction by RNaseT1 (Fig S4A). To confirm that rRNA is not visibly altered in $Smc1\beta^{-/-1a}$ spermatocytes, we also stained wt and mutant spreads with the anti-rRNA antibody Y10b (Fig S4B), showing no difference between the genotypes.

## Loss of SMC1β causes open chromatin and increased transcription near chromosome ends

Higher expression of TERRA would be possible if the chromatin compaction of subtelomeric/telomeric regions is decreased. TERRA itself may contribute to the opening of telomere proximal

chromatin through R-loop formation. More open chromatin would in turn favour more formation of RNA–DNA hybrids. Because we observed both, higher TERRA transcription and RNA–DNA hybrids, we investigated the degree of chromatin accessibility using ATAC-seq. This method probes chromatin compaction by assessing the capacity of a transposable element to insert itself. We performed ATAC-seq on each 5000 FACS-purified 4N spermatocytes from wt, $Smc1\beta^{-/-}$, and $Smc1\beta^{-/-1a}$ spermatocytes. The analysis of ATAC-seq data was performed using the nf-core ATAC-seq pipeline.

We distinguished four regions based on their proximity to chromosome ends: 5, 45, 450, and 4,500 kb from the chromosome ends (Fig 5A–D). Within the most proximal region (5 kb), up to threefold more ATAC-accessible sites were identified in the $Smc1\beta^{-/-1a}$ spermatocytes than in wt. These numbers were also clearly above genome-wide average, which is not much different from the wt signals in the 5-kb region. Most $Smc1\beta^{-/-1a}$ chromosomes showed a 1.5–2fold increase in accessible chromatin. This difference to wt gradually disappeared when the region analyzed was enlarged, reaching further off the chromosome ends. Within the 4,500 kb region, there were very little differences between $Smc1\beta^{-/-1a}$ and the wt, and indeed on many chromosomes, the $Smc1\beta^{-/-1a}$ values

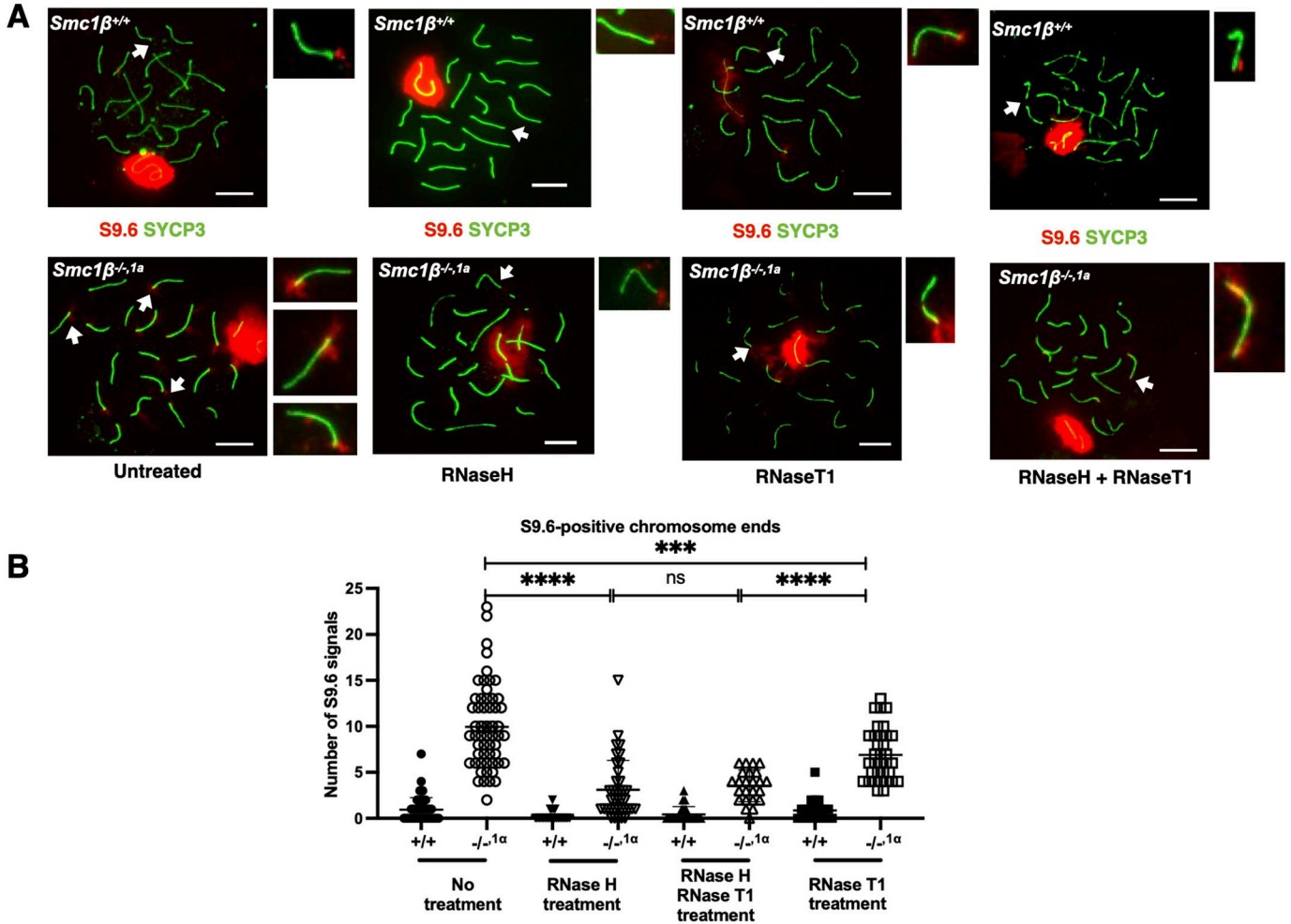

**Figure 4. RNA–DNA hybrid formation in SMC1β-deficient spermatocytes.**
**(A)** DNA–RNA hybrids were detected by the S9.6 antibody (red) used along with anti-SYCP3 for the AEs (green) in $Smc1\beta^{+/+}$ (n = 127) and $Smc1\beta^{-/-,1a}$ (n = 107) spermatocytes. White arrows indicate chromosome areas that are magnified to show in detail S9.6-positive chromosome ends (DNA–RNA hybrids). Shown are spreads untreated, treated with RNaseH, RNaseT1 or both. **(B)** Graph showing the average S9.6 signals at the chromosome ends per cell in $Smc1\beta^{+/+}$ and $Smc1\beta^{-/-,1a}$ spermatocytes. Spreads were treated as indicated. Each data point represents one spermatocyte. Scale bars = 5 $\mu$m. Tukey's multiple comparison test was applied for statistics. Source data are available for this figure.

were even slightly lower (ratio below 1.0) than those of wt chromosomes. Table S1 provides statistical analysis of these data. Analyzing the genomic features of ATAC-seq peaks showed a very similar distribution of ATAC-seq peaks in wt and the two mutants with respect to exons, introns, promoter regions, and other features (Fig S5).

Thus, SMC1β is specifically required to maintain a closed chromatin configuration near the telomeres including at promoter regions, consistent with its role in down-regulating TERRA transcription.

Higher levels of open chromatin near telomeres in SMC1β-deficient spermatocytes may correlate with more transcriptional activity at these regions. RNA-seq using total RNA from FACS-sorted 4N spermatocytes from wt and the two mutants was performed. Fig 6 shows the percentage of total number of transcripts derived from regions in different distances to the chromosome ends. These were 5, 45, 450, and 4,500 kb distance from ends, covering all chromosomes (Fig 6A).

The difference in transcriptional activity between wt and the two mutants near telomeres was much more pronounced than the distance of the telomeres. Up to 150fold more RNA-seq peaks were registered for certain chromosomes in the mutant spermatocytes compared with wt (Fig S6). Overall, the fraction of RNA-seq peaks in the 5-kb telomere-proximal region, reflected app. 4% of the transcripts of wt compared with about threefold higher percentage of transcripts of $Smc1\beta^{-/-}$ (11%) and of $Smc1\beta^{-/-,1a}$ (12%) spermatocytes. This is consistent with increased transcriptional activity within this region or decreased activity elsewhere. The $Wls$ gene, the most telomere proximal gene on chromosome 3 (position bp 159,545,287–159,644,302), is up-regulated 20fold in $Smc1\beta^{-/-1a}$ compared with wt as assessed by qRT–PCR. The increase in transcriptional activity vanished with increasing distance from the chromosome ends. In the 4,500 kb region, there was almost no difference between $Smc1\beta^{-/-1a}$ and wt anymore (Figs 6B and S6). This also shows that there is no down-regulation of telomere-distal

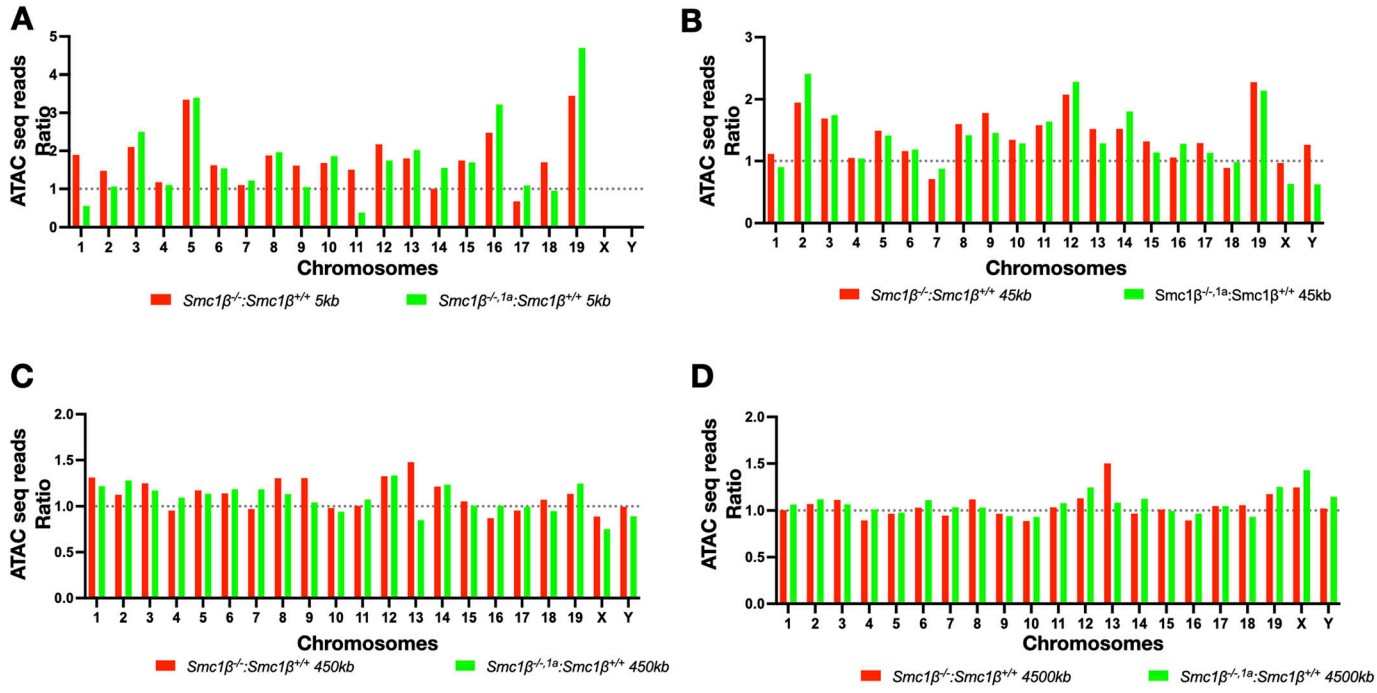

**Figure 5.   ATAC-seq.**
**(A, B, C, D)** Graph showing the fold change distribution of normalized ATAC-seq reads (before peak calling) in the 5 kb sub-telomeric region (A), in the 45 kb sub-telomeric region (B), in the 450 kb sub-telomeric region (C), and in the 4,500 kb sub-telomeric region (D) for each chromosome of the wt and the $Smc1\beta^{-/-}$ and $Smc1\beta^{-/-,1a}$ spermatocytes. Three technical repeats were performed.

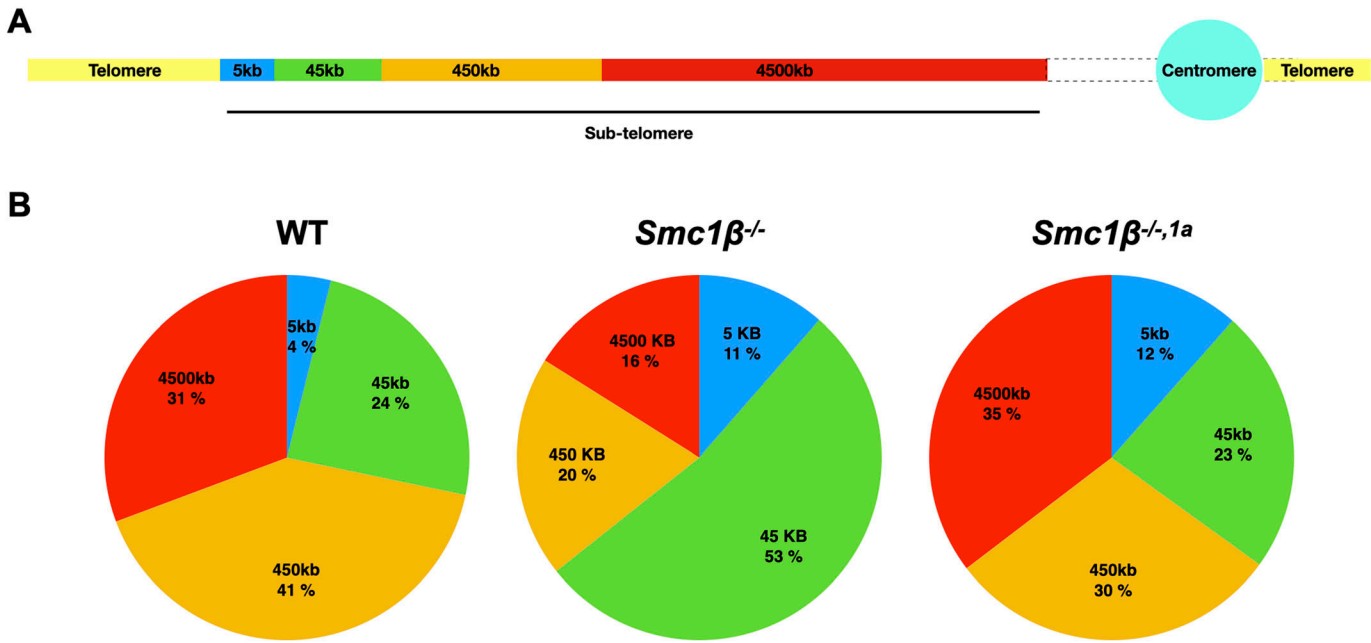

**Figure 6.   RNASeq.**
**(A)** Schematic diagram of a chromosome and its sub-telomeric region showing the regions used for analysis of the total transcripts generated at each of these sub-telomeric regions. **(B)** Pie charts showing the percentages of transcripts, based on the total number of transcripts within 4,500 kb of the sub-telomeric region, for the centromere distal sub-telomeric regions of all chromosomes of wt, $Smc1\beta^{-/-}$, and $Smc1\beta^{-/-,1a}$ spermatocytes, of which, 30,000 cells per mouse from three mice per genotype were sorted.

gene expression in the SMC1β-deficient spermatocytes, which otherwise would have affected the percentages reported above. These data are consistent with the ATAC-seq results and supports the concept of SMC1β as a negative regulator of telomere-proximal transcription in spermatocytes.

# Discussion

The absence of SMC1β from oocytes and spermatocytes was shown to be detrimental for their telomere integrity (Adelfalk et al, 2009), but the underlying reason remained unknown. Here, we show that in spermatocytes, SMC1β is required for suppressing TERRA transcription, that SMC1β restricts transcription in telomeric/subtelomeric regions, that SMC1β promotes closed chromatin in telomeric/subtelomeric regions, and that SMC1β prevents the formation of RNA–DNA hybrid structures in these regions. With these functions, SMC1β is pivotal for establishing a proper chromatin structure and preventing DNA damage in the telomeric/subtelomeric region. This function is unique to SMC1β because, as shown by us before, increased expression of SMC1α at SMC1β-like levels through early diplotene cannot rescue telomere integrity, contrasting several other SMC1β-deficiency phenotypes that are rescued (Biswas et al, 2018). We propose that the opening of chromatin and expression of TERRA with the formation of RNA–DNA hybrid structures, triggered by absence of SMC1β, renders these regions sensitive to DNA damage.

Increased TERRA expression in the absence of SMC1β is much more pronounced in spermatocytes than in oocytes. Although there appeared to be a trend on all chromosomes analyzed towards higher TERRA expression, dictyate-arrested oocytes and embryonic oocytes showed no or only mild statistically significant increase in TERRA. The trend correlates with our earlier report (Adelfalk et al, 2009), where about a fifth of the oocyte chromosomes showed telomere aberrations. Why more telomere aberrations and more strongly increased TERRA expression are seen in spermatocytes is not known. The distinct regulation of the overall gene expression in spermatocytes and oocytes, especially in the long-term arrested adult oocytes, and distinct regulation because of the spermatocyte-specific pachytene checkpoint mechanism and consequences of the unique X/Y sex body may affect TERRA differently in the two types of germ cells.

Like in $Smc1β^{-/-}$ meiocytes (Adelfalk et al, 2009), a lack of synapsis at the telomeres (Bannister et al, 2004) and failed telomere bouquet formation was observed in REC8-deficient cells (Golubovskaya et al, 2006; Hou et al, 2021). Although a systematic analysis of telomere damage in $Rec8$ mutants has still to be reported, this may suggest that a cohesin complex based on SMC3/SMC1β associated with REC8 is responsible for proper telomere dynamics, integrity, and control of TERRA expression. In somatic cells, Rad21 and CTCF depletion by siRNA causes telomere damage loosely reminiscent of that reported for SMC1β deficiency in spermatocytes (Deng et al, 2012a). However, unlike in SMC1β-deficient meiocytes, TRF1, TRF2 are also lost from telomeres in the somatic cells, suggesting a different mechanism such as rendering the shelterin complex unstable or additional indirect effects of the siRNA treatment.

The accumulation of TERRA in SMC1β-deficient spermatoytes to damaged telomeres as opposed to undamaged ones was consistently observed in all types of analyses performed (Fig 3). Although telomere sequences are extended in SMC1β-deficient spermatoytes, the increase of TERRA signals tended to be even higher, suggesting that not only more telomer sequences attract more TERRA, but that the damage of these ends itself correlates with elevated TERRA. Both explanations–more TERRA on damaged chromosome ends and more overall TERRA because of aberrantly lengthened telomeres–are not mutually exclusive. Further interpretation of these observations remains speculative, because from this analysis, one cannot distinguish between TERRA being present before telomeres are damaged or accumulating afterwards on damaged telomeres. Given the reports from somatic cells that increased TERRA expression causes telomere damage (Azzalin et al, 2007; Deng et al, 2012b; Pfeiffer & Lingner, 2012) the former seems likely as our observation of enhanced TERRA on damaged telomeres would be consistent with these reports.

At the ends of several chromosomes of $Smc1β^{-/-}$ spermatocytes, we observed signals through staining with the S9.6 antibody which recognizes RNA–DNA hybrid structures. These signals were sensitive to RNaseH indicating RNA–DNA hybrid structures. The sex body chromatin was also recognized by the S9.6 antibody. Most parts of the sex chromosomes are unsynapsed, that is, each axis consists of only 2 sister chromatids, and here as well, the S9.6 signals were RNaseH sensitive suggesting R-loops. Generally, the use of the synapsis-proficient $Smc1β^{-/-1a}$ model prevented asynapsis-related perturbations of our analysis. Because the S9.6 antibody has recently been reported to also detect rRNA (Smolka et al, 2021), we assessed whether there are changes in rRNA in SMC1β-deficient spermatoytes by use of RNaseT1, which degrades rRNA, and the antibody Y10b, which detects rRNA. RNaseT1 reduced only marginally the S9.6 signal observed in $Smc1β$-deficient spermatocytes, and Y10b signals were not different between the genotypes. Thus, the increase in S9.6 signals seen in $Smc1β^{-/-}$ spermatocytes mainly resulted from RNA–DNA hybrids and correlates with the increased TERRA signals at the ends of chromosomes. We suggest that TERRA-containing RNA–DNA hybrids form at or near the telomers. Whether in the absence of SMC1β these hybrids contribute to telomer damage or are a consequence of an altered telomer structure, or both, cannot be ascertained.

TERRA association with the PAR was described in ES cells (Chu et al, 2017) and is very abundant there. The X/Y chromosomes carry some internal (TTAGGG)n regions, and it remains unclear whether TERRA is transcribed there and remains associated or is derived from telomeres of either the sex chromosomes themselves or the autosomes. In spermatocytes lacking SMC1β, TERRA was very much increased at the PAR, possibly reflecting the special structural features of the spermatocyte sex body which appears to affect cohesin control of TERRA expression.

The increase in transcriptional activity near the chromosome ends does not affect TERRA only but also a number of additional genes close to telomeres that are normally silenced. Whether this contributes to spermatocyte death in the cohesin mutant is unknown. Questions to be addressed in future work concern the mode of recruitment of SMC1β to telomeres and subtelomeric regions, and how SMC1β contributes there to tightly packaging chromatin.

In human cell lines, CTCF was reported to localize to TERRA promoters, but neither the promoters nor near-telomeric CTCF-binding sites nor an actual CTCF localization in this region are known for murine somatic cells or even mouse meiocytes. It may still be tempting to speculate that through an interaction with CTCF or with a similar factor, an SMC1β–cohesin complex supports silencing. Physical interaction between SMC1β and CTCF in pull-down experiments using spermatocyte extracts was not observed by us.

Our findings may also have consequences for human health. It may be interesting to analyze a role for cohesin in somatic cell telomere protection and assess whether this is altered in tumor cells, which often express aberrant patterns of cohesins, sometimes even meiotic cohesins. For reproductive health, based on our observations reported here and described earlier on telomere deficiencies caused by cohesin loss in mouse oocytes (Adelfalk et al, 2009), we suggest that one likely mechanism by which loss of cohesin contributes to age-related chromosome mis-segregation and thus aneuploidy is through deprotection of telomeres. Whereas loss of sister chromatid cohesion is considered a major cause of such mis-segregation (Jessberger, 2012), telomere defects were often correlated to meiotic errors, decreased fertility, and other errors in gametogenesis (Hemann et al, 2001; Liu & Keefe, 2002; Treff et al, 2011; Ferlin et al, 2013; Turner & Hartshorne, 2013), reviewed in Keefe, 2016, 2020). Thus, loss of the key meiotic cohesin SMC1β may have dual effects: deterioration of sister chromatid cohesion and TERRA-associated telomere defects, both contributing to age-related aneuploidy in oocytes.

# Materials and Methods

### Mice

$Smc1β^{-/-}$ mice, Smc1βprom-EGFP, and $Smc1β^{-/-,1a}$ mice were previously described (Revenkova et al, 2004; Adelfalk et al, 2009; Biswas et al, 2018). The mice were maintained according to the guidelines in the animal facility of the Technische Universität Dresden, Medical Faculty. All animal experimentation was approved by the State Animal Welfare Office. The mice were of 6–16 wk of age unless otherwise specified.

### Cell sorting

Seminiferous tubules were isolated from the testes of mice after removal of tunica albuginea. Single-cell suspension was prepared by passing the cells through a 40 $\mu m$ strainer. The cells were collected in PBS and washed one time. Then, the pellet was resuspended in FACS incubation buffer (HBSS supplemented with 20 mM HEPES [pH 7.2], 1.2 mM $MgSO_4$, 1.3 mM $CaCl_2$, 6.6 mM sodium pyruvate, 0.05% lactate, glutamine, and 1% fetal calf serum) at a density of 2 million cells/ml. Then, 5 $\mu g$/ml of Hoechst 33342 was added and cells were incubated for 1 h at 32°C in a water bath. Before sorting, 2 $\mu g$/ml of propidium iodide (PI) is added to exclude the dead cells. Meiocytes were sorted with 85 $\mu m$ in an ARIA II (BD Biosciences). PI and Hoechst were excited using lasers 488 and 355 nm, respectively. 685 LP and 710/40 BP filters were used to detect PI signal. Hoechst blue signal was detected using 600 LP and 620/10 BP filters in front of the first detector and Hoechst red was detected with 440/40 BP filter in front of the second detector. The cells were collected in PBS and frozen at –80°C for subsequent RNA sequencing, qPCR and, protein extraction. For ATAC sequencing, 5,000 cells per genotype were collected. FlowJo (Tree Star Inc.) and FACSDiva software were used for cell sorting and data analysis.

### ATAC sequencing and data analysis

ATAC sequencing sample and library preparation was prepared according to the 10× genomics guidelines. 5,000 spematocytes were sorted into 30 $\mu l$ PBS and transposition was carried in 10 $\mu l$ of reaction buffer. After transposition, DNA was purified either with bead-based methods or with QIAGEN MinElute Kit. Index PCR was performed in 25 $\mu l$ of reaction buffer mixture to prepare the libraries. Libraries were purified twice with 1.3× vol of GE beads. Size distribution was analyzed with a fragment analyzer. Library size distribution was analyzed by the Fragment Analyzer (company) and samples diluted in a 1:3 ratio.

ATAC-seq bioinformatics analysis was done using the nf-core ATAC-seq pipeline (Ewels et al, 2020) with –genome mm10 and default settings otherwise. The pipeline is very comprehensive and automatically runs all required steps, including (but not limited to) raw read QC, adapter trimming, alignment, filtering, estimation of complexity, and generation of normalized coverage tracks. In addition, we ran MACS2 (Zhang et al, 2008) to identify broad and narrow peaks of enriched ATAC-seq signal with the custom parameters -g mm -B -q 0.05 –nomodel –extsize 200 –shift -100 –nolamda –keep-dup -f BAM –call-summits (narrow peaks), and –broad, respectively (broad peaks).

We took individual chromosomes and assigned an annotated telomeric region used in mm10 as the end of the chromosome, which we further used as the total length of the chromosome. Using samtools view command , first we analyzed the number of reads according to the genome region. We measured the number of reads according to the genomic region from the end of the chromosomes. We defined the sub-telomeric region as 5, 45, 450, and 4,500 kb from the chromosome end. The reads per chromosome region is normalized to the region and defined as transcripts per bp (e.g., x reads in 5 kb region; normalized reads as x/5,000). Furthermore, we normalized those reads by genome average reads. Genome average reads are the total number of reads per chromosome length.

### RNA isolation and purification

Total RNA was extracted from sorted spermatocytes and also from total testis with TRIzol reagent (Invitrogen Inc.) according to the manufacturer's guidelines. Total amount of RNA was solubilized in RNAse-free water and quantified using nanodrop for qPCR or tapestation for RNA sequencing.

### RNA sequencing

For library preparation, mRNA was isolated from DNAse-treated total RNA using the NEBNext rRNA depletion Kit (human, mouse, rat) from New England Biolabs (NEB) according to the manufacturer's

instructions. Final elution was done in 5 μl nuclease-free water. The samples were then directly subjected to the workflow for strand-specific RNA-Seq library preparation (NEBNext Ultra II Directional RNA Library Prep; New England Biolabs). For ligation, custom adaptors were used (Adaptor-Oligo 1: 5′-ACA CTC TTT CCC TAC ACG ACG CTC TTC CGA TCT-3′, Adaptor-Oligo 2: 5′-P-GAT CGG AAG AGC ACA CGT CTG AAC TCC AGT CAC-3′). After ligation, the adapters were depleted by XP bead purification (Beckman Coulter) adding beads in a ratio of 1:0.9. Dual indexing was done during the following PCR enrichment (15 cycles, 65°C) using custom amplification primers carrying the index sequence indicated with "NNNNNNN." (Primer1: AAT GAT ACG GCG ACC ACC GAG ATC TAC ACT CTT TCC CTA CAC GAC GCT CTT CCG ATC T, primer2: CAA GCA GAA GAC GGC ATA CGA GAT NNNNNNN GTG ACT GGA GTT CAG ACG TGT GCT CTT CCG ATC T). After two more XP bead purifications (1:0.9), libraries were quantified using the Fragment Analyzer (Agilent). Libraries were equimolarly pooled before sequencing them with a length of 75 bp in single end mode on an Illumina NextSeq 500 system to a depth of at least 40 mio reads.

## RNA-seq data analysis

After sequencing, RNA-SeQC (1.1.8) (DeLuca et al, 2012) was used to perform a basic quality control which includes exonic, intronic, and intergenic distribution of the reads and rRNA rate within each samples. Alignment of the reads to the mouse reference (mm10) was done with GSNAP (v2018-07-04) (Wu & Nacu, 2010) and Ensembl gene annotation version 92 was used to detect splice sites. Here, the standard GSNAP parameters were used for the alignment (Liao et al, 2014).

TPM values were generated with Kallisto (0.43.1) which only needs an index of the transcriptome (Bray et al, 2016). This index was created with the Ensembl gene annotation version 92 and the gtf_to_fasta script from the tophat package. BigWig and Bed files were created for the visualisation of the alignments. Here, the bamCoverage from the package deeptools (3.0.2) was used (Ramirez et al, 2016). Because the region of interest contains repetitive sequences, GSNAP was run with the parameter -n 10 which allows GSNAP to report up to 10 alignments for aligned fragment.

## Quantitative reverse-transcription PCR analysis

RNA was reverse transcribed to cDNA using the Promega cDNA kit (M3682; Promega), together with oligo(dT)$_{15}$ (C1101; Promega). To amplify UUAGGG repeat transcripts, we used CCCTAA-specific oligos for generation of cDNA for qPCR analysis. 1 μg of cDNA was used for qPCR analysis. qPCR was carried out in duplicates using SYBR Green (4309155; Thermo Fisher Scientific). Beta actin was used as a house-keeping gene to normalize the data.

## Analysis of TERRA expression in oocytes

Eight young adult age group female mice (1.5–3 mo) were used for comparing TERRA expression in wt and $Smc1\beta^{-/-}$ mice. Ovaries were collected in M2 media (Merck–Sigma-Aldrich) supplemented with milrinone to prevent maturation. The ovaries were punctured and oocytes were collected and cumulus cells removed. 20 GV stage

oocytes from each mouse were used for RNA isolation. The Picopure RNA isolation kit (Arcturus) was used to isolate RNA and cDNA was prepared by reverse transcribing RNA using oligo dT primers and the Superscript II kit (Invitrogen). Equal volumes of template cDNA (10.5 μl of 25 μl reaction mixture) from oocytes of each mouse were used for measuring TERRA expression by quantitative real time PCR (qPCR). qPCR was done using the Rotor Gene SYBR Green PCR kit on the qTower 2.0 (Analytic Jena). Beta-actin was used as a house-keeping gene for normalization. For TERRA analysis in embryonic oocytes, embryos were obtained at 17.5–18.5 d post coitum. Each female embryo was tailed and genotyped. Ovaries were collected into ice cold PBS, and ovary cell suspensions were prepared by rupturing the follicles. The suspensions were then added to 175 ml of RLT buffer (Quiagen Inc.) and stored at −80°C for further RNA isolation. RNA was isolated as described above using the QIAGEN RNA isolation kit and cDNA was prepared as described above. Equal concentrations (10 ng/ml) of cDNA were used for qPCR using the primers for subtelomeric regions of chromosomes 2,3,5,8 and for the PAR; β-actin was used to normalize the data.

**Primers.**

| b-Actin | Forward: AGGCATTGCTGACAGGATGCAG<br>Reverse: AGCACTTGCGGTGCACGATG |
|---|---|
| Chromosome 2 | Forward: GAGTGCCTTACTATCTCCTAAGTTTT<br>Reverse: TGGAGTTAATTTTGTGGAGGTTG |
| Chromosome 3 (WLS gene) | Forward: CCAGTCTAATGGTGACCTGGG<br>Reverse: TGAGAGTCAGCATGCACCAG |
| Chromosome 5 | Forward: TTGGGTTGGCTGTTCTCGTA<br>Reverse: GAGGCTGAGAGACTTCCGTG |
| Chromosome 8 | Forward: TCCCTCATACTCCGCAGAAC<br>Reverse: GGTGGCTCAGTGGTGAAATG |

## Chromosome spreads

Spermatocyte chromosome spreads were prepared according to the protocol adapted from Wojtasz et al (2009). Tunica albuginea was removed from the testis and single-cell suspension was prepared by passing the cells through an 80-μm strainer. The cells were washed once with PBS and the cell pellet was resuspended in 300 μl of cold PBS. 7 μl of 0.25% of NP40 was added on the slide and mixed with 1.5 μl of single-cell suspension dropwise. The cells were lysed for 2 min and fixed by adding 24 μl of fixation buffer (1% paraformaldehyde, 10 mM sodium borate buffer pH 9.2). Slides were incubated in a closed humidified chamber for 1 h and then dried under the cell culture hood. The slides were then washed twice with photoflo and then washed thrice with water. The slides were then used immediately or kept at −20°C until staining.

## Immunofluorescence staining and FISH

Slides containing chromosome spreads were blocked with 10% goat serum for at least 1 h at room temperature before applying primary antibody. The slides were incubated with primary antibodies for at least 3 h at 37°C or 4°C overnight. Then, the slides were washed with

blocking buffer and incubated with secondary antibodies for at least 1 h at 37°C. After the secondary antibody treatment, the slides were washed with blocking buffer and mounted with Vectashield containing 1 μg/ml of DAPI. Telomeric DNA FISH was performed according to the manufacturer's protocol (Metasystems' GmbH).

### Telomere RNA FISH

Cells were fixed at room temperature for 10 min with 4% para-formaldehyde, and dehydrated with ice cold ethanol series (70%, 85% and 100%) for 5 min each at room temperature. For hybridization cells were incubated overnight at 37°C with 0.5 pmol $(CCCTAA)_5$ oligonucleotide probe in hybridization buffer (2 mg/ml of BSA, 10% dextran sulfate, 5% formamide in 2X SSC buffer). The slides were washed 5 min each with 2X saline-sodium citrate (SSC) once, twice with 50% formamide in 2X SSC at 39°C, two more times at 39°C with 2X SSC, and once with 2X SSC at RT. Nuclei were counterstained with DAPI/Vectashield.

### RNaseH and RNaseT1 treatment and antibody staining of spermatocyte spreads

Spermatocyte chromosome spreads were treated with RNaseT1 (EN0541; Thermo Fisher Scientific), 1:200 dilution in staining buffer PBS with 0.1% Tween supplemented with 3 mM $MgCl_2$ by incubation at 37°C for 1 h (protocol adapted from Smolka et al [2021]). After washing with staining buffer, the spreads were treated with RNaseH in RNaseH Reaction buffer (10 units per 100 μl of reaction mix; M0297L; New England Biolabs) for another 1 h at 37°C. Treatment of spreads with only RNaseT1 (same dilution as mentioned before) and only RNaseH (same concentration as before) was performed for 1 h at 37°C. Subsequent to washing and blocking with 10% goat serum, the spermatocytes were primary immunolabelled with 1:100 dilutions of clone S9.6 mouse monoclonal antibody (MABE1095; Merck Sigma-Aldrich) and rabbit anti-SYCP3 (NB00-230; Novus). Anti-rRNA staining was done with 1:100 dilutions of anti rRNA Y10b (sc-33678; Santa Cruz Biotechnology). In all cases, primary antibody incubation was performed overnight at 4°C; the slides were then washed and stained with 1:1,000 dilution of secondary antibodies (Alexa Fluor 488 goat anti rabbit IgG, A-11034; Invitrogen and Alexa Fluor 568 goat anti mouse IgG, A-11004; Invitrogen). After washing with staining buffer, the slides were mounted with Vectashield containing 1 μg/ml of DAPI.

### Microscopy and image analysis

Fluorescence was visualized with Zeiss Axiophot fluorescence microscope and analysis of images was performed using ImageJ version 1.43u.

### Statistics

Depending on the type of experiments, unpaired nonparametric $t$ test (Mann–Whitney test), unpaired $t$ test, Tukey's multiple comparison test or Dunn's multiple comparison test was used. Graph-Pad prism of version 9.4.0(453) was used for analysis and graph designing.

## Data Availability

RNA-seq and ATAC-seq data are available at the accession number GSE230556.

## Supplementary Information

## Acknowledgements

This study was initially funded through the EU Horizon 2020 grant GermAge to R Jessberger, and then by the Deutsche Forschungsgemeinschaft through grant JE150/30-1 to R Jessberger.

### Author Contributions

U Biswas: conceptualization, data curation, formal analysis, investigation, and writing—review and editing.
T Deb Mallik: data curation, formal analysis, investigation, and writing—review and editing.
J Pschirer: data curation, formal analysis, validation, and investigation.
M Lesche: resources, formal analysis, validation, and writing—review and editing.
K Sameith: resources, data curation, formal analysis, validation, and writing—review and editing.
R Jessberger: conceptualization, resources, formal analysis, supervision, funding acquisition, methodology, project administration, and writing—original draft, review, and editing.

### Conflict of Interest Statement

The authors declare that they have no conflict of interest.

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
