## [Reviewer comments · Life Science Alliance]

Life Science Alliance

Cohesin SMC1 β promotes closed chromatin and controls TERRA expression at spermatocyte telomeres

Rolf Jessberger, Uddipta Biswas, Tanaya Deb Mallik, Johannes Pschirer, Mathias Lesche, and Katrin Sameith

DOI: <https://doi.org/10.26508/lsa.202201798>

Corresponding author(s): Rolf Jessberger, TU Dresden

Review Timeline:

Submission Date:	2022-11-03
Editorial Decision:	2022-12-12
Revision Received:	2023-04-19
Editorial Decision:	2023-04-20
Revision Received:	2023-04-26
Accepted:	2023-04-26

Transaction Report:

December 12, 2022

Re: Life Science Alliance manuscript #LSA-2022-01798-T

Prof. Rolf Jessberger
Dresden University of Technology
Inst. of Physiological Chemistry
Fiedlerstr. 42
Dresden 1307
Germany

Dear Dr. Jessberger,

Thank you for submitting your manuscript entitled "Cohesin SMC1 β promotes closed chromatin and controls TERRA expression at spermatocyte telomeres" to Life Science Alliance. The manuscript was assessed by expert reviewers, whose comments are appended to this letter. We invite you to submit a revised manuscript addressing the Reviewer comments.

Thank you for this interesting contribution to Life Science Alliance. We are looking forward to receiving your revised manuscript.

Sincerely,

B. MANUSCRIPT ORGANIZATION AND FORMATTING:

Reviewer #1 (Comments to the Authors (Required)):

In this study Biswas et al. analyze telomere damage observed in the absence of Smc1b (meiosis-specific cohesion subunit) in mouse germ cells. They had previously shown (Biswas et al., 2018) that expression of Smc1a instead of Smc1b rectifies such defects as reduced axes length and chromosome asynapsis, but not the telomere damage; however telomerase and shelterin proteins protecting telomeres are not affected in the absence of Smc1b. Now the authors claim that Smc1b restricts the expression of a long non-coding RNA TERRA, that is transcribed from the subtelomeric regions and can form R-loops with telomeric repeats. Indeed, in cancer cells TERRA overexpression caused telomeric damage by altering the telomeric structure.

The study explores interesting relationship between structural protein, non-coding RNA and telomeric structure, but contains unsupported conclusions and thus requires a major revision and additional experiments.

General remarks:

1. Study contains gross mistakes in data interpretation on several occasions. For example (page 6): "Investigating oocytes from young adult mice, i.e. oocytes in dictyate arrest, revealed increased TERRA expression for the four chromosomes analyzed, statistically significant for Chromosome 2 and 3 (Suppl. Fig. 2B)"
If statistical tests do not reveal the difference, any statements on increase/decrease should be omitted.
2. The study convincingly demonstrates that TERRA is highly overexpressed (up to 60 times more for some chromosomes) in the absence of Smc1b in Smc1b^{-/-} and Smc1b^{-/-}, 1a spermatocytes; however the putative increase in TERRA localization to the damaged telomeres looks marginal. This apparent discrepancy must be discussed.
3. The number of TERRA signals on telomeres (Fig.2) does not necessarily indicates increased localization of TERRA to telomeres - for example, for Smc1b^{-/-} it rather indicates an increased number of asynapsed chromosome axis. A better parameter would be the total intensity of TERRA signals on telomeres. If the authors do not observe such increase in TERRA signal intensity in the absence of Smc1b, despite a strong increase in TERRA expression, this must be indicated and discussed.
4. Data presentation on TERRA localization on damaged and undamaged telomeres (Fig.3) is extremely confusing. Legend to Fig. 3 refers to statistically significant differences on TERRA localization between WT and mutant cells, but the data is not presented. Fig.3C lacks proper description. Is there statistically significant difference between RNA FISH and DNA FISH shown on Fig.3C? If not, higher TERRA signal on damaged telomeres might reflects a simple increase in the telomere length, rather than preferred localization of TERRA to the damaged telomeres. This possibility needs to be clearly indicated and discussed.
5. The manuscript does not contain evidence that increased expression of TERRA and its putative enhanced localization to the damaged telomeres is directly regulated by Smc1b. Alternative interpretations need to be discussed. To increase the impact, it might be good to show a model illustrating how Smc1b regulates TERRA expression and telomeric structure.

Minor remarks:

1. Figure legends do not indicate what is shown on the graphs as an average (mean, median value?) and what are error bars.
2. It is not clear whether PAR in Fig. 1B is the same as TERRA/mPAR in Suppl. Fig. 2B.
3. No Materials and Methods for Embryonic oocyte collection (Fig. S2B).
4. It is stated (Fig. 2) that, t.ex. RNA FISH was performed on 30 WT cells, but in only 14 cells the TERRA signals were counted. Why were the rest excluded from analysis? The criteria should be described for example, in Materials and Methods.
5. The number of analyzed cells for Fig.2 (14, 11 and 13 cells for WT, SMC1b^{-/-} and Smc1b^{-/-}, 1a, respectively) is too low. How many independent experiments were performed?
6. WT cell in Fig2A is not a good representative, because it displays 41 TERRA signals (on every telomeric end) while the average number of TERRA signals in WT is 27.
7. Fig. 2B shows "Average RNA FISH signal per cell" on y-axis. How averaging was performed?
8. How many cells, and how many experiments were analyzed for Fig.3?
9. RNase treatment had not affected the number of S9.6 signals on chromosome ends in Smc1b^{-/-}, 1a cells (Fig.4D), nor the total intensity of S9.6 signals on autosomes (Fig.S3B). That is in difference with the situation on XY body. What is the reason for the different sensibility of R-loops on XY body and on the telomere ends to the RNase treatment?
10. Fig. 5 does not correspond to its figure legend.
11. What could be the reason for a considerable increase in the number of transcripts from 45kb region in Smc1b^{-/-} in comparison to other samples(Fig.6)?

Reviewer #2 (Comments to the Authors (Required)):

The authors address the question of why SMC1b is required for the protection of telomeres during meiosis. Having previously excluded other mechanisms, they focused their attention on TERRA, a non-coding RNA derived from the sub-telomeric region. They find that the loss of SMC1b results in the ectopic expression of TERRA resulting in increased DNA damage, R-loop formation, and transcriptional activity proximal to the telomers. The authors conclude that SMC1b is required to silence TERRA expression to protect chromosome ends. The authors data support their conclusions, and I am supportive of publication. I have some comments that would need addressing prior.

Major

1. How many replicates were used in the ATAC-Seq analysis?
2. The analysis in figure 5 would need statistical evaluation.

Minor

1. In figure 1b, the error bars in the control are missing.
2. Several variants of RNase H and ATAC-seq were used in the text. Please correct and unify.
3. Figure 4 c and d require statistical evaluation.

Reviewer #3 (Comments to the Authors (Required)):

In this paper, Biswas et al. investigate the effects of loss of the meiosis specific cohesin subunit SMC1-beta on telomere structure and function. It had been observed in previous studies that SMC1-beta deficiency leads to frequent telomere damage events. In the new work it is demonstrated that loss of SMC1-beta leads to a strong increase of TERRA expression from multiple chromosome ends in testis and spermatocytes. The TERRA FISH signal also increased at telomeres in mutant spermatocytes. Damaged telomeres accumulated more TERRA than non-damaged telomeres. R-loops also appeared to accumulate though the signal was not sensitive to RNase H treatment (Figure 4D) rendering this conclusion uncertain. Finally, ATACSeq analysis demonstrates increased accessibility of telomeres in SMC1-beta deficient cells coinciding with increased RNA transcripts near chromosome ends. Overall, the paper documents interesting structural and functional changes at damaged telomeres that arise from SMC1-beta deficiency. Two scenarios seem possible. TERRA may be preferentially recruited to the damaged telomeres in order to promote an appropriate telomeric DNA damage response. Alternatively, TERRA and TERRA R-loops may cause the damage in the first place. I consider the findings of this paper interesting but the R-loop experiments should be cleaned up before publication.

Comments:

Figure 4D: The S9.6 signal in SMC1-beta cells clearly increases over wt. However, since the signal is not sensitive to RNaseH it is unclear if R-loops do increase as postulated. The S9.6 antibody also recognized rRNA in the absence of R-loops (see <https://doi.org/10.1083/jcb.202004079>). Therefore, appropriate controls are crucial to be certain about the R-loops. If the microscopy analysis remains ambiguous, the authors may consider carrying out DRIP experiments with the essential RNaseH controls.

The cited paper by Lopez de Silanes et al. (NatComm 2014) which states that mouse TERRA stems mainly from chromosome 18 has been disproven in the current paper as well as publications by multiple other laboratories. See e.g. (doi: 10.1261/rna.076281.120 ; doi:10.1093/nar/gkx958 ; doi:10.1093/nar/gky1149 ; doi: 10.1093/biolre/ioy243). Thus, I would not cite the flawed Lopez de Silanes paper.

Response to Referees

We thank the referees for their insightful comments which helped improving the manuscript.

We are detailing the changes and providing our point-by-point responses below. The most important changes to the paper are:

- addition of new data on TERRA RNA FISH, Figure 2;
- addition of new data on oocyte TERRA signals (Figure S2) with many more cells analyzed and an additional chromosome included, which improved the statistics on all samples;
- eliminating a major confusion with Figure 3 and the corresponding legend;
- including a new Figure 4 with new data showing RNaseT1 and RNaseH treatment of spermatocyte spreads, and a new supplemental figure showing Y10b rRNA staining
- added statistical evaluation of the ATAC-Seq data (new Suppl. Table 1)

With this and the other, more minor changes to the paper we believe to have more than fulfilled the reviewers' requests.

Thank you
Rolf Jessberger

Referee #1

1. For Supplemental Figure 2A and B, embryonic and adult oocytes, many more oocytes were analyzed to improve the statistics and critically assess the potential increase of TERRA. Also one more chromosome was analyzed for embryonic oocytes (A). With this new data, it turned out that while there is a trend towards higher TERRA expression on all ten chromosomes tested (no exception) in the mutant, only one of the five chromosomes of embryonic oocytes showed a moderately significant increase in TERRA in the *Smc1b*^{-/-} oocytes. For the adult oocytes, similarly a significant increase in TERRA was determined only for one of five chromosomes. Thus, we concluded that oocytes show no or little increased TERRA in both stages, embryonic and adult. The difference between spermatocytes and oocytes is discussed in the Discussion section.

2. Yes, the increase in TERRA signals specifically on damaged telomers compared to undamaged telomers is smaller than the overall increase of TERRA on all telomers in *Smc1b*^{-/-} cells compared to the wt (Fig. 3). And that is to be expected. It seems that we did not make sufficiently clear that the analysis of the TERRA signals on damaged versus undamaged telomers was done within the *Smc1b*-deficient cells. This was not a comparison between wt and *Smc1b*^{-/-} cells. The comparison in Fig. 1 between the two genotypes shows a massive increase in TERRA expression for the mutant. Comparing different telomers within the mutant, meaning within overall high levels of TERRA, is an entirely different approach and the result is very interesting in that there is uneven distribution of TERRA between undamaged and damaged telomers, i.e. two- to threefold more TERRA on damaged telomers within the TERRA-high mutant cells. We are describing this more clearly now both in the Results section, the legend, and the Discussion.

The confusion mentioned in this referee's point No. 4 originates indeed from our failure to make clear that we are not comparing wt and mutant but telomers within the mutant. We have clarified this. The wt DNA and RNA FISH is shown only now in the supplement (S3) solely to illustrate this type of staining for the wt in comparison to the mutant.

3. We agree and have followed the referee's advice to quantify the total TERRA signal intensity Figure 2C. The figure was further improved by better insertion of clearer images, which show one cell in each instance (2A), and by adding many more data points (2B, C). The statistical significance is very clear.

4. Please see above, #2.

In addition we are now discussing the two – not mutually exclusive – explanations for higher TERRA signals on damaged telomers, i.e. preferential localization of TERRA to damaged telomers or increased TERRA signals due to extended telomers. This is now discussed in the Discussion section.

The previous figure legend was indeed confusing and we apologize for this. We think this has now been clarified.

5. The referee is correct in writing that we have no direct evidence how SMC1b controls TERRA expression (directly/indirectly). Such evidence would be extremely difficult to obtain in this system (ChIP would not suffice). Therefore, we refrain from presenting a model which would be overly suggestive of a mechanism.

Minor Points:

1. We are providing information on the statistical method applied in each legend where statistical analysis is shown.

2. We have clarified this and now the labels are consistent. The PAR (Fig. 1B, Fig. S2) are indeed the same.

3. The M&M section on embryonic oocyte preparation and analysis has been added; sorry for omitting this initially.

4. & 5. As we have added many more data points, this was resolved.

6. We have improved Fig. 2 and are showing now a much better representative image for the WT.

7. Our previous legend to Fig. 2B unfortunately was unclear. We are showing the number of TERRA RNA FISH signals per cell. We have now analyzed more than 60 cells for each genotype and are showing the distribution and the mean. In the newly added Fig. 2C, the average of the total TERRA FISH signals per cell is provided. We think the legend now describes this properly.

8. We are providing the cell number now for DNA and RNA FISH in the legend. It was 45 and 56 cells.

9. The chromatin environment of the XY chromosome is very different from that of autosomes. The XY chromosomes contain relatively more RNA-DNA hybrids and may thus respond more drastically to RNase treatment. There is limited knowledge of RNA-DNA hybrids in spermatocytes and to further explore this specific issue is clearly beyond the scope of this paper, but consistently we observed RNase treatment to reduce the S9.6 signal on XY, consistent with the autosomal data. Please note that figure 4 has been substantially improved by including a new control, the treatment with RNaseT1 to

eliminate potential background of S9.6 staining from rRNA, and by adding more data points (127, 107 cells resp.).

10. The legend now fits the figure.

11. We speculate that SMC1b-cohesin contributes to heterochromatinization of the chromosome ends, the subtelomeric and telomeric regions. Silencing close to the ends but gradually loosening this silencing and allowing more open chromatin to exist the further distal from the end the region is located as there are more and more genes located.

Referee #2

1. We performed 3 technical repeats of the ATAC-Seq experiment and have now noted this in the legend.

2. We have performed statistical evaluation of all chromosomes for all three genotypes for all four regions, i.e. different distances from the telomers. The further we go from the telomer the less statistical significant differences between wt and the two mutants are observed. We have put the data into a table and added this to the supplement (Supplemental Table 1).

Minor points

1. We have improved Figure 1 by addition of error bars 1B. Except of course for the Smc1b RNA expression since there is none in the Smc1b^{-/-} spermatocytes.

2. We have corrected the spelling of RNase and ATACSeq.

3. We have redesigned Figure 4 and have added many more data points, including a new control using RNase T1. The statistics is now clearly shown in Fig. 4B.

Referee #3

1. Figure 4: We thank the referee for pointing out the potential problems using the widely used S9.6 antibody. In the paper cited by the referee, several types of transformed cell lines are shown and staining by S9.6 is widespread in the cytoplasm and the nucleus. We do not see that widespread staining. Rather we specifically observed staining of the sex body and the chromosome ends, and we analyzed quantitatively specifically the signals at the chromosome ends. Thus, these cannot be nucleolar, rRNA-derived signals. Therefore we are confident not to report the artefacts described in Smolka et al., 2021.

Nevertheless, we have now included new data using RNaseH and RNaseT1 to distinguish between RNA-DNA Hybrids and rRNA. The data presented in a new Figure 4 and Supplemental Figure 4 show that indeed there is an increase in RNA-DNA Hybrids. T1 only slightly reduces the signals, much less than RNaseH treatment (Fig. 4B).

We also used the anti rRNA antibody Y10b for an additional control. This showed that Y10b signals are equally present in wt and mutant and are both equally reduced by RNase T1 treatment. Thus, there is no difference rRNA presence as detected by this antibody.

We have eliminated the paper by de Slianes et al., 2014.

April 20, 2023

RE: Life Science Alliance Manuscript #LSA-2022-01798-TR

Prof. Rolf Jessberger
TU Dresden
Inst. of Physiological Chemistry
Fiedlerstr. 42
Dresden 1307
Germany

Dear Dr. Jessberger,

Thank you for submitting your revised manuscript entitled "Cohesin SMC1 β promotes closed chromatin and controls TERRA expression at spermatocyte telomeres". We would be happy to publish your paper in Life Science Alliance pending final revisions necessary to meet our formatting guidelines.

- please add your supplementary figure legends to the main manuscript text
- please add the Twitter handle of your host institute/organization as well as your own or/and one of the authors in our system
- please add the author contributions and a conflict of interest statement to the main manuscript text
- please use the [10 author names, et al.] format in your references (i.e. limit the author names to the first 10)
- please include a Data Availability statement indicating accession information for the RNAseq and ATACseq data

Figure Check:

- please add scale bars to Figure 3 A,B; Figure S3 A,B

A. FINAL FILES:

B. MANUSCRIPT ORGANIZATION AND FORMATTING:

Thank you for your attention to these final processing requirements. Please revise and format the manuscript and upload materials within 3 days.

Sincerely,

April 26, 2023

RE: Life Science Alliance Manuscript #LSA-2022-01798-TRR

Prof. Rolf Jessberger
TU Dresden
Inst. of Physiological Chemistry
Fiedlerstr. 42
Dresden 1307
Germany

Dear Dr. Jessberger,

Thank you for submitting your Research Article entitled "Cohesin SMC1 β promotes closed chromatin and controls TERRA expression at spermatocyte telomeres". It is a pleasure to let you know that your manuscript is now accepted for publication in Life Science Alliance. Congratulations on this interesting work.

DISTRIBUTION OF MATERIALS:

Again, congratulations on a very nice paper. I hope you found the review process to be constructive and are pleased with how the manuscript was handled editorially. We look forward to future exciting submissions from your lab.

Sincerely,
